# The Pathological and Histopathological Findings in Cats with Clinically Recognised Hypertrophic Cardiomyopathy Are Related to the Severity of Clinical Signs and Disease Duration

**DOI:** 10.3390/ani15050703

**Published:** 2025-02-27

**Authors:** Izabela Janus-Ziółkowska, Joanna Bubak, Rafał Ciaputa, Małgorzata Kandefer-Gola, Agnieszka Noszczyk-Nowak

**Affiliations:** 1Department of Pathology, Wrocław University of Environmental and Life Sciences, CK Norwida 31, 50-375 Wrocław, Poland; joanna.bubak@upwr.edu.pl (J.B.); rafal.ciaputa@upwr.edu.pl (R.C.); malgorzata.kandefer-gola@upwr.edu.pl (M.K.-G.); 2Department of Internal Medicine and Clinic of Diseases of Horses, Dogs and Cats, Wrocław University of Environmental and Life Sciences, Grunwaldzki Sq 47, 50-366 Wrocław, Poland

**Keywords:** feline, heart, microscopy, postmortem examination, pathology, HCM

## Abstract

Feline cardiomyopathies are important cardiac diseases leading to severe clinical signs and often sudden death. The most common form of cardiomyopathy in cats is hypertrophic cardiomyopathy (HCM). Despite its frequency, the diagnosis is challenging and requires ruling out other causes of cardiac hypertrophy. The postmortem confirmation of diagnosis is not performed frequently. Our study aimed to describe the pathological and histopathological pattern in cats with clinically stated HCM. The retrospective analysis was performed on 34 hearts subjected to postmortem examination and included detailed gross morphological and histopathological investigation. Our results show that the pathological pattern of HCM in cats is very diverse, with more severe changes noted in cats diagnosed at a younger age and treated for a longer time. The study points to a further need for cooperation between clinicians and pathologists to broaden our knowledge on HCM pathology, possible causes, and outcomes.

## 1. Introduction

Cardiomyopathies are the most common form of feline cardiac disease, including hypertrophic cardiomyopathy (HCM), dilated cardiomyopathy (DCM), restrictive cardiomyopathy (RCM), and arrhythmogenic right ventricular cardiomyopathy (ARVC) [1,2,3,4]. Recently, left ventricular noncompaction (LVNC) has also been described in cats [3,5]. In ambiguous cases, the term “nonspecific phenotype cardiomyopathy (NCM)” is used [1,3,4,6].

The diagnosis of hypertrophic cardiomyopathy requires echocardiographic confirmation of left ventricular wall thickening and the elimination of other causes that can lead to similar structural and/or functional changes, e.g., congenital cardiac defects, hormonal disorders, or systemic hypertension. Nonetheless, drawing a clear line between primary and secondary changes may sometimes be challenging [1,2,3,4,6,7,8].

Several gene mutations responsible for the development of cardiomyopathies have been recognised so far; nonetheless, the causes of almost all cardiomyopathies remain unsolved [1,2,4,9,10,11,12]. It is an open question if HCM should be understood as a uniform disease or rather as a collection of numerous subtypes of the disease, all with left ventricular thickening, but with various known or unknown causes, evolution over time, and the ability to respond to treatment and prognosis [2].

The postmortem confirmation of the diagnosis is not frequently performed, primarily due to the lack of owner consent and costs. Moreover, the histopathological diagnosis of HCM in cats remains equivocal [8,13]. Histopathological changes include cardiomyocyte disarray, enlargement and degeneration, medial hypertrophy of the walls of small coronary arteries, but may also include mild interstitial fibrosis or inflammatory infiltration [2,8,14,15,16,17]. Nonetheless, some cats with a clinical diagnosis of HCM and left ventricular thickening may also not exhibit cardiomyocyte hypertrophy or hyperplasia [2,16]. Due to its complex nature, human medicine recommends the involvement of an experienced cardiac pathologist in the examination of cases with cardiac hypertrophy [18].

We hypothesise that despite ruling out other apparent clinical causes of left ventricular hypertrophy, HCM remains an ambiguous diagnosis with a diverse histopathological pattern. Therefore, the aim of the study was to present the pathological and histopathological characteristics of cats clinically diagnosed with HCM.

## 2. Materials and Methods

Patient records of cats subjected to postmortem examination in the Cardiopathology Unit between 2018 and 2023 were retrospectively analysed to identify animals with an HCM diagnosis. The clinical diagnosis and treatment were performed in accordance with available guidelines [4] (Appendix A) by various experienced veterinary cardiologists cooperating with the Cardiopathology Unit. The history, clinical signs, diagnosis, and survival time were available in the cover letter sent with the sample by the clinician and archived with the postmortem examination record. The inclusion criteria were (a) a clinical diagnosis of HCM based on echocardiographic examination [4] and (b) confirmation of left ventricular hypertrophy during the postmortem examination. The exclusion criteria were congenital cardiac disorders, a history of hyperthyroidism, systemic hypertension, diabetes mellitus, chronic kidney disease, and neoplastic disease (cardiac and extracardiac).

The available clinical history was analysed, and information about clinical signs (auscultation disorders [muffled heart sounds, heart murmur, gallop rhythm, tachycardia, arrhythmia], respiratory distress, pulmonary oedema, pleural effusion, pericardial effusion, and aortic thromboembolism [ATE]) was noted. The clinical signs were scored as follows: each of the categories was scored 0 if the symptom was absent and 1 if it was present in the patient’s history. A total score for each animal was counted. Moreover, the survival time from the moment of diagnosis (the patient’s first echocardiographic examination showing left ventricular thickening) to death or euthanasia was retrieved. The survival time served as a criterion to divide animals into two categories: short survival group (S group; ≤1 week of treatment) and long survival group (L group; >1 week of treatment).

Hearts were collected from the animals during necropsy. All animal owners gave their consent for postmortem and histopathological examination. According to local ethics regulations, no further consent was required as the examination was part of a diagnostic process.

During the necropsy, hearts were dissected from the respiratory tract, leaving 1 cm of aorta and pulmonary artery. The pericardial sac was removed, and the heart was flushed of blood clots and fixed in 7% buffered formalin for 24 h. After fixation, a detailed pathomorphological examination was carried out by a pathologist experienced in cardiac pathology (IJZ), as described previously [19]. Briefly, the external examination included heart weight, height (largest longitudinal diameter), and width (largest transverse diameter) of the whole heart and left atrial appendage. Heart weight exceeding 20 g and heart weight-to-body weight ratio exceeding 4 g/kg served as confirmation of cardiac hypertrophy [2]. Next, the hearts were sectioned transversely at the level of the upper third of the ventricular height, and the thickness of the left ventricular wall and interventricular septum were measured. All measurements were taken using a manual calliper with an accuracy of 0.1 mm. After the gross examination, specimens of the left ventricular wall, right ventricular wall, and interventricular septum at transverse section (at the level of the upper third of the ventricular height) and longitudinal section (at the level of the cardiac apex) [19], and left and right atrium and atrial appendages were obtained from each heart. The specimens were embedded in paraffin blocks, cut into 6 μm sections, stained using haematoxylin-eosin (HE) and Picrosirius red stains [20], and underwent detailed microscopic examination. The microscopic evaluation included cardiomyocyte degeneration, cardiomyocyte hypertrophy, myocardial disarray (evaluated only in the left ventricular and interventricular septum specimens), endocardial thickening and fibrosis, myocardial fibrosis, myocardial fatty infiltration, and myocardial inflammatory infiltration. All histopathological features were assessed and scored as described previously [19,21,22] and presented in Table 1.

For the purpose of the analysis performed in the study, for every patient, a histopathological score for each cardiac wall (left ventricular wall, right ventricular wall, interventricular septum, left atrial wall, right atrial wall) was calculated as a sum of scores for histopathological changes in the subjected wall (Figure 1A). Moreover, for each patient, a degeneration score, hypertrophy score, endocardial thickening score, fibrosis score, fatty infiltration score, and inflammation score were calculated by summing the scores for each feature from all cardiac walls (Figure 1B). Finally, a generalised histopathological score was calculated for each patient as a sum of all histopathological changes.

The statistical analysis was performed using Statistica 13.3 software (Tibco, Cracow, Poland) and appropriate tests. Data normality was tested using the Shapiro–Wilk analysis. The difference between groups was obtained using either the Mann–Whitney U test or the Kruskal–Wallis analysis with the Dunn post-hoc test. The difference between sex distribution was assessed using the chi-square test. Correlation analysis was performed using Spearman’s correlation test. We correlated clinical history results (age at the moment of diagnosis, age at the moment of death or euthanasia, body weight, and total clinical score) with the gross pathological measurements (heart weight, heart weight-to-body weight ratio, LAA height and width, left ventricular posterior wall thickness, and interventricular wall thickness) and the histopathological results. Additionally, we correlated the gross pathological measurements with the histopathological results. The significance was set at *p* ≤ 0.05.

## 3. Results

Based on the inclusion and exclusion criteria, thirty-four animals were enrolled in the study. The group consisted of fifteen domestic shorthairs, seven British shorthairs, four Maine Coons, two Sphynx cats, two Scottish folds, and one cat of each of the following breeds: British longhair, Neva Masquerade, Siberian, and Persian. The median age was 7 years (range 1–18 years), and the median weight was 5 kg (range 3.7–10 kg). The initial diagnosis was made at a median age of 6 years (ranged 6 months to 18 years). Twenty-five animals were male. The population study for S and L groups is presented in Table 2. Group S showed a significantly higher age of initial diagnosis compared to the L group (*p* = 0.01).

Twelve cats (L group) were diagnosed with hypertrophic cardiomyopathy and treated according to ACVIM guidelines [4] for at least four months prior to euthanasia (*n* = 10) or sudden death (*n* = 2). One cat in that group (case #31) was presented for a screening examination (without clinical signs); the diagnosis of stage B1 HCM was made [4] and the animal suddenly died after a year before the follow-up examination. Twenty-two animals (S group) presented with sudden onset heart failure and/or ATE, were diagnosed and stabilised but the decision for euthanasia was made (*n* = 18) or sudden death occurred (*n* = 4) in less than a week from the initial diagnosis.

The clinical history of the examined cats is shown in Appendix A. Due to a retrospective nature of the study and the availability of only the information presented in cover letters, the exact left ventricular measurements were accessible only in 11 cases, and left atrial measurements in 15 cats. In the remaining patients, information about echocardiographically stated concentric left ventricular hypertrophy and left atrial enlargement was noted in the record without the exact measurements.

The most common clinical symptom was respiratory distress reported by the owners (*n* = 23), followed by sudden ATE (*n* = 15), pleural effusion (*n* = 13) and pulmonary oedema (*n* = 11). Other clinical signs were reported in less than 20% of cats each (Appendix A). The S group presented more frequently with ATE than the L group (*p* = 0.045). Other clinical signs separately and the total clinical score showed no difference between the groups (*p* > 0.05). In six animals, sudden death was noted, including four animals in advanced heart failure with survival time of less than a week (S group) and two animals diagnosed four months and one year prior, respectively (L group).

During the postmortem examination, cardiac congenital defects and structural disorders (e.g., aortic stenosis, mitral valve dysplasia) were ruled out. Cardiac hypertrophy was confirmed in all examined cases. The median heart weight was 32 g (range 22–52 g), and the median heart weight-to-body weight ratio was 6.7 g/kg (range 4.1–10.8 g/kg). Groups S and L showed no difference in heart weight and heart weight-to-body weight ratio (*p* = 0.34 and *p* = 0.51, respectively).

The detailed results of the gross examination are presented in Appendix A. Unfortunately, due to the retrospective analysis of the study, the left atrial appendage dimensions were not available in one case. The results of the gross examination in groups S and L are presented in Table 3.

All cats showed mild to severe enlargement of the left atrial appendage (Figure 2).

The left atrial appendage was significantly wider in S than in in the L group (*p* = 0.009; Table 3). The left atrial appendage height and width strongly correlated with cardiac weight (*p* = 0.0006; r = 0.75; and *p* < 0.0001; r = 0.86, respectively) and moderately correlated with the heart weight-to-body weight ratio (*p* = 0.04; r = 0.58; and *p* = 0.02; r = 0.63, respectively).

The median interventricular septum thickness was 9.0 mm (range: 5.8–11.6 mm) and the left ventricular wall thickness was 9.6 mm (range: 5.7–13.5 mm). The left ventricle and interventricular septum showed no differences between the S and L groups (*p* > 0.05; Table 3). Although all animals presented with either symmetrical or asymmetrical left ventricular thickening, the histopathological picture of the population was very diverse (Appendix A; Figure 3).

Only 10 cases showed clearly typical histopathological signs of HCM: cardiomyocytes hypertrophy, degeneration, and myocardial disarray (Figure 3B). Additionally, five cats presented with fatty infiltration in the right ventricular wall (Figure 3C,D; including two cases with concomitant inflammatory infiltration). Nine cases presented with extensive myocardial fibrosis (Figure 3E,F; including three cases with visible thickening and fibrosis of the ventricular endocardium and five cases accompanied by inflammatory infiltration). Seventeen cases presented with inflammatory infiltrates of various extents (Appendix A). In the majority of cases (*n* = 13) lymphocytic infiltrates were noted, followed by mixed infiltrates composed of lymphocytes and plasma cells, neutrophils and lymphocytes, or neutrophils, lymphocytes, and plasma cells. In three cases, the inflammatory infiltration was found only in one cardiac wall, while in other cases, multiple cardiac parts were involved. The inflammation had various extents: from single inflammatory cells dispersed within the myocardium (*n* = 6), through multiple inflammatory cells dispersed within the myocardium (*n* = 4), to multiple groups of inflammatory cells in the myocardial tissue (*n* = 7). Although it was not possible to clearly distinguish myocarditis with secondary cardiomyocyte lysis from myocardial necrosis with secondary inflammatory infiltration, in seven cases, myocarditis without substantial fibrosis was suspected (Figure 3G,H; affecting ventricles and/or atria). One case (#18; Figure 3I,J) presented with a short clinical history and atypical histopathological picture with cardiomyocyte hypertrophy, fibrosis and vacuolization of the myocardium, suggestive of NCM, or atypical HCM.

The most visible histopathological change was cardiomyocyte degeneration, followed by cardiomyocyte hypertrophy, myocardial fibrosis, and fatty infiltration (Appendix A; Table 4).

As compared with the L group, the S group presented with (Table 5): less intense left ventricular wall fibrosis (*p* = 0.04), interventricular septum fibrosis (*p* = 0.04), right ventricular wall degeneration (*p* = 0.01), left ventricular wall histopathological score (*p* = 0.0002), interventricular septum histopathological score (*p* = 0.004), and total histopathological score (*p* = 0.003).

In the study population, the left atrial appendage width correlated with the left ventricular histopathological score (*p* = 0.006; r = 0.47) and the total histopathological score (*p* = 0.01; r = 0.42). Moreover, the left ventricular posterior wall thickness negatively correlated with the left ventricular fibrosis score (*p* = 0.04; r = −0.63) and the left ventricular endocardial thickening and fibrosis score (*p* = 0.006; r = −0.77).

The histopathological changes were present not only in the ventricular specimens but also in the atrial specimens in all animals (Appendix A; Table 5), including 31 cases presenting changes in both the left and right atrium. The left atrium presented with various patterns, including cardiomyocyte degeneration and hypertrophy, myocardial fibrosis, and inflammatory infiltrates (*n* = 27). In five cases, only cardiomyocyte degeneration was present in the left atrium. On the contrary, in the majority of cats (*n* = 18), the changes in the right atrium were visible only as cardiomyocyte degeneration of various intensities. In 16 animals, the picture was more complex, with fibrosis, inflammatory infiltration, or fatty replacement. The atrial changes did not differ between the S and L groups (Table 5). The left atrial degeneration correlated with the left atrial appendage height (*p* = 0.01; r = 0.43) and width (*p* = 0.002; r = 0.52). Moreover, the left atrial histopathological score positively correlated with the left atrial appendage width (*p* = 0.007; r = 0.46).

Although present in all parts of the heart, histopathological changes were mostly visible in the left ventricle, followed by the interventricular septum. The left ventricle showed a higher histopathological score as compared to the left atrium (*p* = 0.002), the right atrium (*p* < 0.0001), and the right ventricular wall (*p* < 0.0001). The interventricular septum showed a higher histopathological score as compared to the right ventricular wall (*p* = 0.0006), the right atrial wall (*p* < 0.0001), and the left atrial wall (*p* = 0.03). Moreover, the left atrium showed a higher histopathological score as compared to the right atrium (*p* = 0.048).

The patient’s age at diagnosis was negatively correlated with left ventricular cardiomyocyte degeneration (*p* = 0.0005; r = −0.58), right ventricular cardiomyocyte degeneration (*p* = 0.002; r = −0.53), left atrial endocardial thickening and fibrosis (*p* = 0.02; r = −0.41), left ventricular histopathological score (*p* = 0.01; r = −0.45), degeneration score (*p* = 0.04; r = −0.36), disarray score (*p* = 0.04; r = −0.36), and endocardial thickening and fibrosis score (*p* = 0.02; r = −0.4).

The severity of clinical signs correlated positively with the left ventricular inflammation score (*p* = 0.04; r = 0.36), the interventricular septum inflammation score (*p* = 0.02; r = 0.4), and the total inflammation score (*p* = 0.04; r = 0.36). Moreover, the clinical symptom score showed a correlation with the values of left atrial degeneration (*p* = 0.045; r = 0.35), left atrial cardiomyocyte hypertrophy (*p* = 0.03; r = 0.36), and the total left atrial histopathological score (*p* = 0.01; r = 0.43).

## 4. Discussion

The retrospective study demonstrates that the histopathological features of cats diagnosed with hypertrophic cardiomyopathy are complex, affecting not only the ventricles but also the atria.

While HCM in Maine Coon, Ragdoll, and Sphynx cats may be caused by a genetic mutation, the disease is most commonly observed in domestic shorthair cats [2,6]. Affected cats typically show signs at an average age of 6.8 years, although Ferasin et al. [6] reported a wide age range from 0.5 to 16 years with equal distribution between males and females. Our study indicates that a diagnosis at a younger age is associated with longer survival but more advanced histopathological changes at the time of death or euthanasia, especially cardiomyocyte degeneration, myocardial disarray, endocardial thickening, and fibrosis, mainly in the left ventricle and left atrium. Cats enrolled in the study presented typical signs of HCM, including respiratory distress/tachypnoea, heart murmur or gallop sound, pulmonary oedema, pleural effusion, and ATE [2,4]. The most severe sign of cardiomyopathy is sudden death as a result of rhythm disturbances or thromboembolic disease (including nervous system thromboembolus) [2,3,4,6,23]. In our study, the presence of ATE was related to shorter animal survival: In 22 animals, the clinical onset led to sudden death or to the decision of euthanasia in less than one week, and 13 cats in that group showed ATE. Moreover, a more severe clinical picture was related to a higher risk of inflammatory changes in the myocardium (especially in the left ventricular wall and interventricular septum) and more severe atrial histopathological changes (especially cardiomyocyte degeneration and hypertrophy). Unfortunately, the ECG examination results were not available; therefore, the presence of rhythm disturbances and their impact on disease course and myocardial remodelling could not be analysed.

Left ventricular hypertrophy can be caused by concurrent HCM and another disorder (e.g., hyperthyroidism, systemic hypertension), leading to a more complex clinical course of the disease and ambiguous postmortem results [2,19,22]. In the current study, we excluded all cats that were clinically diagnosed with hormonal disturbances or systemic hypertension (either primary or secondary) to avoid confusion about whether the cardiac changes are induced by secondary disease or primary HCM. Another group excluded from the study was comprised of cats with chronic kidney disease (CKD), regardless of the systemic blood pressure history. Systemic hypertension, although not always present in CKD, is very common in that disease [22,24], and the mechanism behind the association between the two diseases is still unclear [24]. A recent study by Flora et al. [22] showed that old cats show similar histopathological changes in the hearts (especially myocardial disarray, cardiomyocyte hypertrophy, microvascular changes, and myocardial fibrosis), disregarding the presence or absence of systemic hypertension, and that approximately three-fourths of animals (in both groups) had signs of CKD. In light of that finding, we decided to exclude cats with a diagnosis of CKD even if they did not present with elevated blood pressure to avoid bias.

Changes in left ventricle led to an increase in left atrial pressure, causing left atrial enlargement [2,23]. In 14 cases, the left atrial enlargement was confirmed by precise echocardiographic measurements. As the left atrium is a complex structure and obtaining reliable measurements postmortem is challenging, we used the left atrial appendage size, as proposed previously [19]. In the current study, the appendage height and width were enlarged in all cases, as compared to values reported in the literature for healthy cats [19], and this was related to an increase in heart weight and heart weight-to-body weight ratio. Additionally, the left atrial appendage showed higher diameters with increased histopathological changes in the left ventricle and in the whole heart. Apart from the atrial enlargement, changes in the histopathological structure were also visible, with histopathological changes advancing with appendage enlargement. It is probable that the structural lesions result not only from the atrial enlargement and tissue stretch but are also related to the primary myocardial disease, similarly to dogs with DCM [21,25].

The postmortem confirmation of the diagnosis of HCM (especially in cases of sudden death) requires not only the thickening of the left ventricular wall and/or interventricular septum but also an increase in heart weight [2,8,15]. A heart weight of 20 g is considered a cut-off point. Moreover, cats with HCM present with a higher heart weight-to-body weight ratio as compared to healthy animals (average 6.3–6.4 g/kg vs. 3–4 g/kg) [2,8]. It is important to remember that after death, the myocardium contracts; therefore, the measured wall dimensions will be higher than clinically obtained measurements for diastolic thickness of the left ventricular wall and interventricular septum [2], but cats with HCM present with both interventricular septum thickness and left ventricular wall thickness higher than in control cats [8]. In our study, all cats presented with a cardiac weight higher than 20 g, a heart weight-to-body weight ratio higher than 4 g/kg, and mean interventricular septum thickness and left ventricular wall thickness higher than in healthy cats reported in the literature [8].

The histopathological pattern of HCM is varied, including myocardial degeneration, hypertrophy, and disarray, along with the signs of local inflammatory response and fibrosis [2,8,13,14,15,16,17]. In a study conducted by Khor et al. [26], cats with pre-clinical HCM showed cardiomyocyte hypertrophy together with the presence of inflammatory infiltrates. At the same time, Fonfara et al. [27] noted that active inflammation is not involved in the structural cardiac remodelling in the advanced stage of HCM. Similarly, our results are ambiguous, with a high percentage of cats presenting with myocardial fibrosis and myocardial inflammatory infiltrates.

In the course of HCM, the myocardium is replaced with fibrous tissue as a result of ongoing myocyte damage and death, which was also observed in our study with a higher prevalence of myocardial fibrosis in animals treated for a longer time. The fibrosis causes myocardial diastolic dysfunction similar to that observed in RCM [2,8,28]. In both situations, the increased stiffness of the left ventricle will eventually lead to left atrial enlargement and signs of heart failure [2,3,4]. It is established that myocardial fibrosis in HCM is accompanied by left ventricular thickening, while in RCM, the thickness of the wall is normal [3,4,6,29,30]. The classification is challenging, as progressing fibrosis may lead to an eventual reduction in left ventricular wall thickness, which was also observed in our study. The classification also does not explain questionable cases, e.g., myocardial thickening with fibrosis but without cardiomyocyte hypertrophy, myocardial thickening accompanied by endocardial (or endomyocardial) fibrosis and thickening (typical for the endomyocardial form of RCM), or cats with sudden onset clinical signs (including sudden death) and severe myocardial fibrosis. In the current research, we found 10 cases of such animals.

In addition to HCM, transient myocardial thickening (TMT) is diagnosed as a thickening of the left ventricle that resolves spontaneously over time. Changes in the left ventricle are accompanied by left atrial enlargement and heart failure [2,31,32,33]. In the cases presented in the literature, left ventricular hypertrophy and left atrial enlargement are not as severe as in HCM cats; nonetheless, a substantial overlap exists [2,31,32,33]. Although it is not a commonly noted disorder, it may lead to sudden onset clinical signs and/or death. TMT has been previously described in a group of 21 cats [31] and in two case reports [32,33]. In none of the papers was a necropsy and histopathological examination performed, as the animals’ echocardiographic measurements returned to normal and the cats survived the disease. Therefore, it is impossible to rule out TMT in cases of cats euthanised due to an HCM diagnosis and sudden onset heart failure. As one of the described cats with TMT was *Bartonella henselae*-positive [32] and two were *Toxoplasma gondii*-positive [33] and returned to normal after antibacterial or antiparasitic treatment, another possible cause underlying TMT is a treatable myocarditis. In none of the cats in the current study was a recurrence of normal cardiac dimensions observed; therefore, TMT was not diagnosed.

It is believed that feline myocarditis is a rare condition [3]. In our study, 17 cats showed inflammatory infiltration in the left ventricle and/or other parts of the heart. In that group, seven animals showed moderate to severe inflammatory infiltration, suggestive of myocarditis. A previous study in dogs [34] showed that myocarditis can be associated with various clinical pictures, including DCM- and HCM-like phenotypes. Additionally, in cats, inflammation of the cardiac wall can resemble various types of cardiomyopathies, including HCM, RCM, and DCM [3,33,35,36]. In the current study, the inflammatory infiltrates were mainly lymphocytic, lymphoplasmacytic, or in seldom cases mixed with neutrophils; the intensity of inflammation ranged from mild (with single small foci of inflammatory cells dispersed within the myocardium) to severe (with vast myocardial areas replaced by inflammatory infiltration). The hearts were moderately hypertrophied, with a normal to moderately thickened interventricular septum, a mild to moderately thickened left ventricular wall, and a severely enlarged left atrial appendage, as compared to healthy cats [19].

Another cause of left ventricular thickening that needs to be taken into account is neoplastic disease, including lymphoma or the presence of metastatic foci [2,37,38]. In the current study, cats diagnosed with left ventricular thickening with the concurrent presence of neoplastic infiltration in other organs were excluded in the initial phase. In none of the remaining cats were neoplastic cells visualized within cardiac specimens.

In the current study, we noted five cats showing not only left ventricular thickening but also fatty replacement in the right ventricular wall. The observed histological image of the right ventricle is typical for ARVC [3]; nonetheless, there is no available research showing concomitant ARVC with left ventricular thickening in cats. In our opinion, the topic requires further exploration.

The most important limitation of the study results from its retrospective nature. The clinical history was available only as a cover letter archived together with the postmortem examination record. It was reported in a short manner; therefore, some important relationships between clinical and pathological patterns could be missed.

Another serious limitation of the study was the relatively high number of cats that presented sudden onset clinical signs without a previous history of echocardiographic examination. The screening examination is common among purebred cat owners; nonetheless, not many mixed-breed or domestic shorthair cat owners are aware of the possibility and need for periodic examination. The availability of year-by-year clinical history would enrich the data available for analysis and, as a result, expand our knowledge of disease course, progression, and pathological outcome.

Another limitation is the lack of complete echocardiographic measurements in multiple cases resulting from the retrospective nature of the study and, in the majority of cats, from a severe clinical condition that allowed only focused cardiac ultrasonography [39]. To circumvent that limitation, a thorough postmortem cardiac examination was performed. Increased heart weight, thickening of left ventricular walls, and enlargement of left atrial appendages served as confirmation of left ventricular hypertrophy and left atrial enlargement.

Moreover, the information regarding ECG examination results was not available in the pathological records; therefore, we cannot recognise the rhythm disturbances as an additional cause of clinical signs or structural changes in the atrial and ventricular myocardium.

## 5. Conclusions

The current study shows that cats diagnosed with HCM, with clinical exclusion of other causes of myocardial hypertrophy, show a diverse and vast histopathological pattern with more severe histopathological changes related to longer treatment duration. Despite ruling out other causes of left ventricular thickening, pathological confirmation of HCM is challenging, with a high percentage of ambiguous cases. It remains unclear if cats diagnosed with HCM present one disease with multiple variants or rather separate conditions.

Our study highlights the need for further research that would combine detailed year-by-year clinical histories and postmortem examination (pathological and histopathological) to expand our knowledge of the disease course and clinicopathological correlations. The complex pathogenesis of ventricular hypertrophy (especially in older animals with the presence of comorbidities) complicates the choice of an appropriate study group, which can lead to overinterpretations.

## Figures and Tables

**Figure 1 animals-15-00703-f001:**
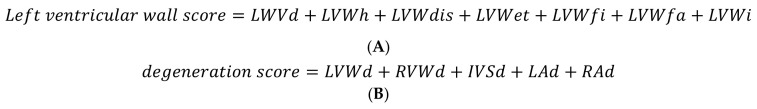
Example of the scoring system used in the study. (**A**). Method of calculating the histopathological score for the left ventricular wall; the same method was used for other cardiac walls. (**B**) Method of calculating the score for degeneration; the same method was used for other histopathological features. LVW—left ventricular wall; d—degeneration; h—hypertrophy; dis—disarray; et—endocardial thickening; fi—fibrosis; fa—fatty infiltration; i—inflammation; RVW—right ventricular wall; IVS—interventricular septum; LA—left atrium; RA—right atrium.

**Figure 2 animals-15-00703-f002:**
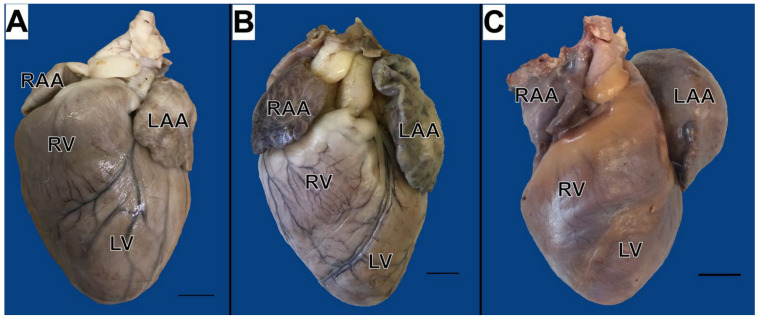
The pathological examination of cats with a clinical diagnosis of hypertrophic cardiomyopathy (HCM) showing the enlargement of the left atrial appendage. Cranial aspect of the heart. Formalin-fixed organs. (**A**) Mild enlargement (case #31). (**B**) Moderate enlargement (case #19). (**C**) Severe enlargement (case #15). Scale bar indicates 1 cm. LAA—left atrial appendage; LV—left ventricle; RAA—right atrial appendage; RV—right ventricle.

**Figure 3 animals-15-00703-f003:**
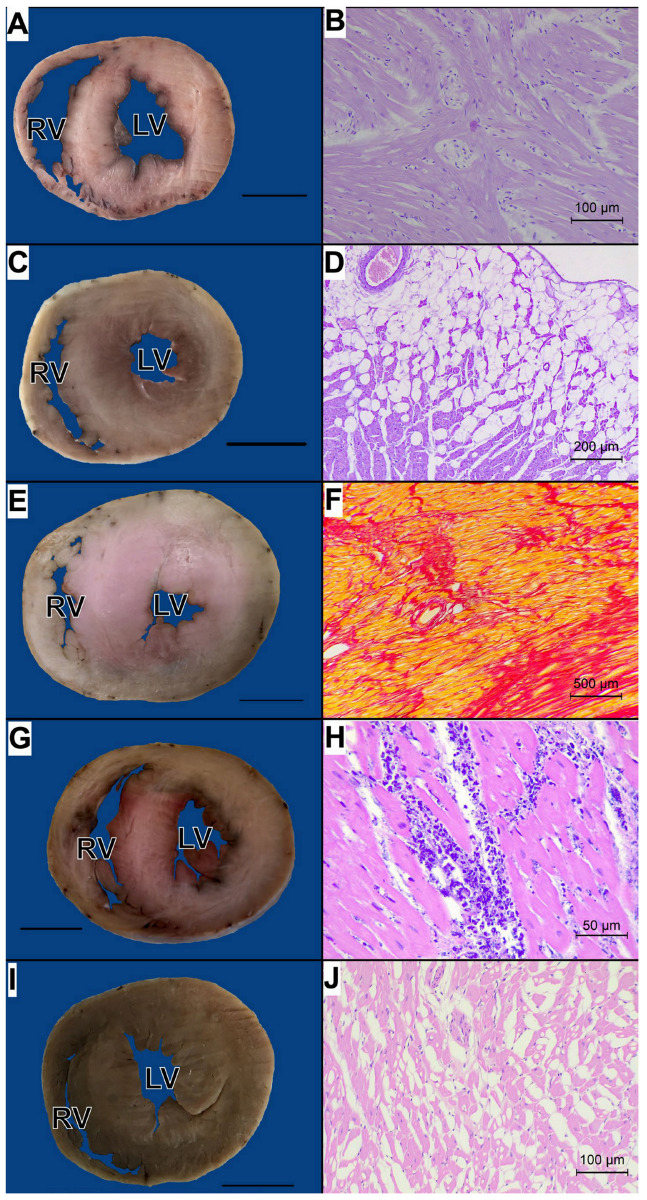
The pathological presentation of cats clinically diagnosed with hypertrophic cardiomyopathy (HCM). (**A**) Cross-section of the ventricles in a case with histological signs of HCM (case #15); bar indicates 1 cm. (**B**) Histopathological image of HCM (case #15): cardiomyocyte degeneration and hypertrophy, and myocardial disarray is visible in the left ventricle; HE stain. 200×. (**C**) Cross-section of the ventricles in a case with fatty infiltration of the right ventricle (case #13); scale bar indicates 1 cm. (**D**) Histopathological image of a case diagnosed with the presence of fatty infiltration in the right ventricle (case #13): cardiomyocytes in the right ventricle are replaced with fat tissue; epicardium top right; HE stain. 100×. (**E**) Cross-section of the ventricles in a case diagnosed with severe myocardial fibrosis (case #31); scale bar indicates 1 cm. (**F**) Histopathological image of a case diagnosed with severe myocardial fibrosis (case #31): severe interstitial and replacement fibrosis (red) of the left ventricular cardiac muscle; Picrosirius red stain. 40×. (**G**) Cross-section of the ventricles in a case diagnosed with the presence of severe inflammatory infiltrates (case #34); scale bar indicates 1 cm. (**H**) Histopathological image of a case diagnosed with the presence of inflammatory infiltrates (case #34): a large focus of myocardial lysis with mixed neutrophilic-lymphocytic inflammatory infiltration between cardiomyocytes; HE stain. 400×. (**I**) Cross-section of the ventricles in a case diagnosed with atypical HCM or nonspecific phenotype cardiomyopathy (NCM) (case #18); scale bar indicates 1 cm. (**J**) Histopathological image of a case diagnosed with NCM (case #18): cardiomyocytes contain empty vacuoles, leading to a spongy appearance of the myocardium; HE stain. 200×. RV—right ventricle; LV—left ventricle.

**Table 1 animals-15-00703-t001:** The criteria used in histopathological evaluation and scoring [19,21,22].

	0	1	2	3
Cardiomyocyte degeneration	Normal myocardium	Subtle cardiomyocyte changes: loss of cross-striation	Moderate cardiomyocyte changes: cardiomyocyte swelling and cytoplasmic degeneration	Severe cardiomyocyte changes: hyperchromatosis and loss of cell structure
Cardiomyocyte hypertrophy	Normal myocardium	Enlarged fibres with large, open, pleomorphic vesicular nuclei and prominent nucleoli in <25% of specimens	Enlarged fibres with large, open, pleomorphic vesicular nuclei and prominent nucleoli in 25–50% of specimens	Enlarged fibres with large, open, pleomorphic vesicular nuclei and prominent nucleoli in >50% of specimens
Myocardial disarray *	Absent	Present		
Endocardial thickening and fibrosis	Absent or thickening extends 1–2 cardiomyocyte cell layers in <50% of the endocardium	Thickening extends 3–4 cardiomyocyte cell layers in <50% of the endocardium	Thickening extends 3–4 cardiomyocyte cell layers in >50% of the endocardium	Thickening extends 4+ cardiomyocyte cell layers in >50% of the endocardium
Fatty infiltration	No or single foci per slide	Present in 10–25% of slide	Present in 25–50% of slide	Present in >50% of slide
Inflammatory infiltration	No inflammatory cells	Single inflammatory cell dispersed in the myocardium	Multiple inflammatory cells dispersed in the myocardium	Multiple groups of inflammatory cells in the myocardium

* Feature was evaluated only in the specimens of the left ventricular free wall and interventricular septum.

**Table 2 animals-15-00703-t002:** The population study and clinical score in the short survival (S) and long survival (L) groups evaluated in the research.

	S Group *n* = 22	L Group *n* = 12	*p*-Value
Age of death/euthanasia [y] median (min–max)	7.5 (1–18)	5.5 (1–15)	0.15
Age of initial diagnosis [y] median (min–max)	7.5 (1–18)	4 (0.5–13)	**0.01**
Weight [kg] median (min–max)	5.0 (3.7–9.0)	5.0 (3.8–10.0)	0.41
Sex distribution M:F	15:7	10:2	0.48
Total clinical score	2 (1–4)	2 (1–5)	0.28

Results are presented as median (range) for age, weight, and total clinical score, and as a male/female ratio for sex distribution. A statistically significant *p*-value is marked in bold.

**Table 3 animals-15-00703-t003:** The results of the gross pathological examination of the hearts of cats with short (S group) and long (L group) survival.

	S Group *n* = 22	L Group *n* = 12	*p*-Value
Heart height [mm]	55.3 (39.3–63.5)	45.9 (33.0–59.6)	**0.02**
Heart width [mm]	36.4 (29.0–46.0)	33.6 (25.4–45.5)	0.20
LAA height [mm]	24.4 (13.5–38.0)	20.5 (11.8–36.4)	0.26
LAA width [mm]	24.2 (17.4–34.2)	18.7 (12.6–26.5)	**0.009**
IVSt [mm]	9.2 (5.8–11.6)	8.7 (6–11.3)	0.34
LVPWt [mm]	10.1 (5.7–13.0)	9.2 (7.7–13.5)	0.33

Results are presented as median (range). Statistically significant *p*-values are marked in bold. LAA—left atrial appendage; IVSt—interventricular septum thickness; LVPWt—left ventricular posterior wall thickness.

**Table 4 animals-15-00703-t004:** The histopathological pattern of the examined animals in the study population.

Histopathological Feature	Histopathological Score Median (Range)
Myocardial degeneration	9 (0–15) ^1,2,3,4,5^
Cardiomyocyte hypertrophy	1 (0–15) ^1,6,7^
Myocardial fibrosis	3 (0–10) ^2,8,9^
Endocardial thickening and fibrosis	0 (0–6) ^3,6,8^
Fatty infiltration	0 (0–9) ^4,7,9,10^
Inflammatory infiltration	1 (0–9) ^5,10^

The histopathological score was calculated as the sum of scores for each cardiac wall and could range from 0 to 15. Statistically significant differences are marked with superscript: ^1^
*p* < 0.0001; ^2^
*p* < 0.0001; ^3^
*p* < 0.0001; ^4^
*p* < 0.0001; ^5^
*p* < 0.0001; ^6^
*p* = 0.046; ^7^
*p* = 0.01; ^8^
*p* = 0.0004; ^9^
*p* < 0.0001; ^10^
*p* = 0.01.

**Table 5 animals-15-00703-t005:** The histopathological score in animals with short (S group) and long (L group) survival.

Histopathological Feature	S Group *n* = 22	L Group *n* = 12	*p*-Value
Left ventricular wall score	4 (0–8)	7 (5–10)	**0.0002**
LVWd	2 (0–3)	3 (2–3)	0.07
LVWh	0 (0–3)	1 (0–3)	0.32
LVWdis	0 (0–1)	0 (0–1)	0.81
LVWet	0 (0–0)	0 (0–3)	0.24
LVWfi	1 (0–2)	2 (0–3)	**0.04**
LVWfa	0 (0–2)	0 (0–0)	0.67
LVWi	0 (0–3)	0 (0–3)	0.54
Interventricular septum score	4 (0–8)	7 (3–10)	**0.004**
IVSd	2 (0–3)	3 (1–3)	0.07
IVSh	0 (0–3)	1 (0–3)	0.34
IVSdis	0 (0–1)	0 (0–1)	0.93
IVSet	0 (0–0)	0 (0–3)	0.44
IVSfi	0 (0–2)	2 (0–3)	**0.04**
IVSfa	0 (0–1)	0 (0–0)	0.84
IVSi	0 (0–3)	0 (0–3)	0.91
Right ventricular wall score	2 (0–5)	4 (1–10)	0.83
RVWd	1 (0–3)	2 (1–3)	**0.01**
RVWh	0 (0–1)	0 (0–3)	0.15
RVWet	0 (0–3)	0 (0–3)	0.87
RVWfi	0 (0–2)	1 (0–2)	0.07
RVWfa	0 (0–3)	0 (0–0)	0.29
RVWi	0 (0–2)	0 (0–3)	0.99
Left atrial wall score	3 (0–8)	4 (2–12)	0.66
LAd	2 (0–3)	3 (0–3)	0.25
LAh	0 (0–0)	0 (0–3)	0.12
LAet	0 (0–1)	0 (0–2)	0.44
LAfi	0 (0–3)	1 (0–3)	0.40
LAfa	0 (0–3)	0 (0–0)	0.53
LAi	0 (0–3)	0 (0–3)	0.99
Right atrial wall score	2 (1–6)	2 (0–6)	0.66
RAd	1 (0–3)	2 (0–3)	0.73
RAh	0 (0–0)	0 (0–3)	0.71
RAet	0 (0–1)	0 (0–0)	0.84
RAfi	0 (0–3)	0 (0–1)	0.96
RAfa	0 (0–3)	0 (0–0)	0.40
RAi	0 (0–2)	0 (0–1)	0.76
Total score	16 (8–31)	25 (14–42)	**0.003**

The results are presented as median (range). LVW—left ventricular wall; IVS—interventricular septum; RVW—right ventricular wall; LA—left atrium; RA—right atrium; d—cardiomyocyte degeneration; h—cardiomyocyte hypertrophy; dis—myocardial disarray; et—endocardial thickening and fibrosis; fi—fibrosis; fa—fatty infiltration; i—inflammatory infiltration; statistically significant *p*-values are marked in bold.

## Data Availability

The data used in the study are available within the manuscript and Appendix A.

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
