# Peer review of "The Pathological and Histopathological Findings in Cats with Clinically Recognised Hypertrophic Cardiomyopathy Are Related to the Severity of Clinical Signs and Disease Duration"

_animals, 2025, doi:10.3390/ani15050703_

Round 1
Reviewer 1 Report
Comments and Suggestions for Authors
The authors present a study on the pathological and histopathological findings in cats with clinically recognized hypertrophic cardiomyopathy, which are related to the severity of symptoms and disease duration.
The study is very interesting, well-written, and highly detailed in terms of the methodology used. It attempts to categorize the different variables in a logical manner. The methods are well-described and easily reproducible. The results are compelling and may have a significant impact on the clinical community.
I would suggest a few minor changes:
Consider using the pathologist's name as an abbreviation (e.g., initials).
Signs and symptoms are different: A symptom is a manifestation of disease apparent to the patient himself, while a sign is a manifestation of disease that the physician perceives. The sign is objective evidence of disease; a symptom is subjective. Please correct this in the title and throughout the text.
For Picrosirius red stains, include a bibliographic reference (e.g., a textbook).
Which inflammatory cells were observed? Were they categorized independently?
Add some comments on the inflammatory infiltrate, its significance, and the distinction between necrosis and myocarditis.
Congratulations on the article!
Author Response
Dear Reviewer,
Thank you very much for taking the time to review this manuscript. Please find the detailed responses below and the corresponding revisions/corrections highlighted red in the re-submitted files.
The authors present a study on the pathological and histopathological findings in cats with clinically recognized hypertrophic cardiomyopathy, which are related to the severity of symptoms and disease duration.
The study is very interesting, well-written, and highly detailed in terms of the methodology used. It attempts to categorize the different variables in a logical manner. The methods are well-described and easily reproducible. The results are compelling and may have a significant impact on the clinical community.
Thank you for that comment and appreciation of our work.
I would suggest a few minor changes:
Consider using the pathologist's name as an abbreviation (e.g., initials).
We added the pathologist’s initials in the Materials and Methods section (line 112)
Signs and symptoms are different: A symptom is a manifestation of disease apparent to the patient himself, while a sign is a manifestation of disease that the physician perceives. The sign is objective evidence of disease; a symptom is subjective. Please correct this in the title and throughout the text.
Thank you for that comment and explaining the difference. The term “symptoms” was replaced with “signs” throughout the manuscript and in supplementary files.
For Picrosirius red stains, include a bibliographic reference (e.g., a textbook).
We added an appropriate reference.
Which inflammatory cells were observed? Were they categorized independently?
The types of observed inflammatory cells are described in Supplementary Table 3. We also added additional information in the Results section (lines 256-266)
Add some comments on the inflammatory infiltrate, its significance, and the distinction between necrosis and myocarditis.
We provided additional information, as suggested (lines 256-266). According to our experience, as the inflammation itself may cause myocardial lysis and necrosis, in some cases it is difficult to state if we observe myocardial necrosis with secondary inflammatory infiltration or if we observe myocarditis with secondary myocardial lysis. But, as we already mentioned in our manuscript (lines 266-268), in five cases myocarditis without substantial fibrosis was suspected rather than myocardial necrosis with secondary inflammation.
Congratulations on the article!
Thank you very much!
Reviewer 2 Report
Comments and Suggestions for Authors
Authors present a study entitled “The pathological and histopathological findings in cats with clinically recognised hypertrophic cardiomyopathy are related to the severity of symptoms and disease duration.” This study adds very good information related to feline cardiology and I believe it is well structured and written. However, I have some suggestions to improve the value of this paper, which are listed below:
Line 61: I suggest adding word “or” before systemic hypertension
Line 84: I suggest using “between 2018 and 2023” rather than “in years….”
Line 134: Figure 1A and B seems to be missing, or the formula is presented as figure?
Also, regarding the formulas (A and B) it is no clear whether the formulas are developed specifically for this study, or are these already available formulas in the literature? Could authors clarify this?
Line 153: Chi square test?
Line 153: could authors please specify what was correlated with what using the Spearman correlation?
Line 163 and elsewhere: it is not necessary to state which test was used; the value of P is enough.
Line 187: authors state that: “Group AD more frequently presented with ATE than group CD (p=0.045; Mann-Whitney U analysis).” And also in the MM section they state that the score was calculated. Based on this, I believe that mean values of the total clinical score should be provided.
Line 194: It is known that cardiac wall becomes thicker after fixation. Moreover, an ex-vivo heart is not under the pressure of a living animal. Can author explain how did they diagnose hypertrophy?
Line 219: I am not sure whether this analysis does add any information. I would expect a larger LAA to increase in all diameters.
Figure 3: I suggest flipping images E, G and I with the RV towards the left side of the image for consistency.
Line 251: It is unclear to me how figure 3A showed cardiomyocytes hypertrophy, degeneration and myocardial disarray since that is a macroscopic photo
Lines 271-282: This paragraph repeats the data from table 4. I suggest avoiding repeating the information.
Line 338-340 I do not understand whether this was a correlation or a comparison. Based on the test it seems there is a correlation, therefore, authors should present it as such. I believe that there was an inverse proportionality between the BW and clinical score.
Overall, I believe that authors performed too many correlations that somehow dilute the core of the study and is becoming a little confusing. For example I would exclude the correlations from lines 299 – 304. I do not see how LAA would negatively correlate with fibrosis in the fibrosis. Also, at line 338 authors perform correlations between clinical score and BW which in my opinion is not within the study’s aim. I suggest rechecking carefully which correlations would be important and could explain some hypothesis and worth keeping rather than performing so many correlations and again, diluting the most important finding of this study.
Supplementary file: Table 1: I recommend changing LA/Ao > 1.5 with word enlarged. Since the clinicians stated in some cats that the LA is enlarged it does not mean necessarily that the LA/Ao was above 1.5. Moreover, some (actually many) clinicians assess the LA in long axis (antero-posterior) diameter rather than LA/Ao. Therefore, if the clinicians did not use the formulation > 1.5 I would avoid stating this in the table.
Author Response
Dear Reviewer,
Thank you very much for taking the time to review this manuscript. Please find the detailed responses below and the corresponding revisions/corrections highlighted red in the re-submitted files.
Authors present a study entitled “The pathological and histopathological findings in cats with clinically recognised hypertrophic cardiomyopathy are related to the severity of symptoms and disease duration.” This study adds very good information related to feline cardiology and I believe it is well structured and written.
Thank you for that comment and appreciation of our work.
However, I have some suggestions to improve the value of this paper, which are listed below:
Line 61: I suggest adding word “or” before systemic hypertension
Correction made as suggested.
Line 84: I suggest using “between 2018 and 2023” rather than “in years….”
Correction made as suggested.
Line 134: Figure 1A and B seems to be missing, or the formula is presented as figure?
Also, regarding the formulas (A and B) it is no clear whether the formulas are developed specifically for this study, or are these already available formulas in the literature? Could authors clarify this?
The formula is presented as a figure. The formula were developed for the study - we added appropriate information in the Materials and Methods section (line 134).
Line 153: Chi square test?
Yes, we meant chi square test.
Line 153: could authors please specify what was correlated with what using the Spearman correlation?
We added information in the Materials and methods section (lines 156-161)
Line 163 and elsewhere: it is not necessary to state which test was used; the value of P is enough.
Thank you for that comment. In our previous papers, multiple times we had comments from the reviewers to specify the type of statistical test together with the p-value. We agree that it is not necessary if is well explained in the Materials and Methods section. We removed the names of the tests from results.
Line 187: authors state that: “Group AD more frequently presented with ATE than group CD (p=0.045; Mann-Whitney U analysis).” And also in the MM section they state that the score was calculated. Based on this, I believe that mean values of the total clinical score should be provided.
Thank you for that comment. We added the mean values of clinical signs score to Table 2. We did not add the the values for ATE as it was scored 0 or 1 as other clinical signs. The details on the presence of ATE in separate cases is provided in Supplementary Table 1.
Line 194: It is known that cardiac wall becomes thicker after fixation. Moreover, an ex-vivo heart is not under the pressure of a living animal. Can author explain how did they diagnose hypertrophy?
After death the cardiac wall gets thicker than end-diastolic in vivo measurement but it does not change with fixation for short periods of time (to 72h): DOI:10.3390/vetsci12010074. As there are not established normal values for feline cardiac dimensions (i.e. left ventricular posterior wall thickness and interventricular septum thickness), we did not evaluate increase in diameters. We used increase in weight, what is an established method in veterinary medicine and used 20g as a cut-off value of heart weight and simultaneously 4 g/kg as a cut-off value for heart weight-to-body weight ratio. We added an appropriate reference in the Materials and methods section (lines 114-115) to the explanation that was already in the Discussion section (lines 378-390). The distinction between cardiomegaly (increase in diameteres) and cardiac hypertrophy (increase in cardiac weight) is according to recommendations in human cardiac pathology (DOI: 10.1007/s00428-021-03038-0)
Line 219: I am not sure whether this analysis does add any information. I would expect a larger LAA to increase in all diameters.
Thank you for that comment. We removed the sentence about correlation of LAA height and width.
Figure 3: I suggest flipping images E, G and I with the RV towards the left side of the image for consistency.
The figure was changed, as suggested.
Line 251: It is unclear to me how figure 3A showed cardiomyocytes hypertrophy, degeneration and myocardial disarray since that is a macroscopic photo
Thank you for that comment. The reference to Figure 3A-B could be misleading and was changed to Figure 3B.
Lines 271-282: This paragraph repeats the data from table 4. I suggest avoiding repeating the information.
Thank you for that comment. We removed the paragraph, as suggested.
Line 338-340 I do not understand whether this was a correlation or a comparison. Based on the test it seems there is a correlation, therefore, authors should present it as such. I believe that there was an inverse proportionality between the BW and clinical score.
It was a correlation. Nonetheless, in the light of the next comment, we removed the correlation between clinical score and BW.
Overall, I believe that authors performed too many correlations that somehow dilute the core of the study and is becoming a little confusing. For example I would exclude the correlations from lines 299 – 304. I do not see how LAA would negatively correlate with fibrosis in the fibrosis. Also, at line 338 authors perform correlations between clinical score and BW which in my opinion is not within the study’s aim. I suggest rechecking carefully which correlations would be important and could explain some hypothesis and worth keeping rather than performing so many correlations and again, diluting the most important finding of this study.
Thank you for that comment. We revised the Results section, removed the unnecessary correlations, left the ones that we believe are meaningful and in the scope of the study and discussed them in more detail (lines 335-336, 344-345 and 371-373). We left the correlation of LAA width with left ventricular histopathological results, as we believe that left atrial enlargement is caused related not only to left ventricular thickening (grossly) but also to left ventricular histopathological changes. Also, we left the sentence about a negative correlation between LVPW thickness and left ventricular fibrosis and endocardial fibrosis as it is important for the topic of distinction between RCM and long-lasting HCM (discussed in lines 399-412).
Supplementary file: Table 1: I recommend changing LA/Ao > 1.5 with word enlarged. Since the clinicians stated in some cats that the LA is enlarged it does not mean necessarily that the LA/Ao was above 1.5. Moreover, some (actually many) clinicians assess the LA in long axis (antero-posterior) diameter rather than LA/Ao. Therefore, if the clinicians did not use the formulation > 1.5 I would avoid stating this in the table.
Thank you for that comment. We changed Supplementary Table 1, as suggested.
Reviewer 3 Report
Comments and Suggestions for Authors
Comments to Author:
Manuscript ID: animals-3476636-peer-review-v1 entitled "The pathological and histopathological findings in cats with clinically recognised hypertrophic cardiomyopathy are related to the severity of symptoms and disease duration".
This manuscript describes the pathological and histopathological findings in cats with clinical hypertrophic cardiomyopathy. I appreciate the authors for conducting this study, and I believe the information provided in their work contributes to the professional knowledge of feline cardiomyopathy. Here are some suggestions and comments for the authors to consider:
Line 91-94: Regarding the exclusion criteria, could you please explain the reasoning behind excluding cases that had received corticosteroid administration? Additionally, if a cat had chronic kidney disease but no systemic hypertension, would it also have been excluded? Please clarify the concern behind this criterion (perhaps in the discussion section).
Line 95, Line 97, Line 98 (and throughout the manuscript): Replace symptom with clinical signs to align with standard veterinary terminology.
Line 96: What does ‘auscultation disorders’ mean? Please clarify this term.
Line 96 (and throughout the manuscript): Replace dyspnea with respiratory distress for proper veterinary terminology.
Line 96 (and throughout the manuscript): Replace hydrothorax with pleural effusion
Line 96 (and throughout the manuscript): Replace hydropericardium with pericardial effusion
Line 99-100: If the duration data were retrieved from the period between echocardiographic diagnosis and death (including euthanasia), I suggest rephrasing "disease duration" to "survival time" for better clarity.
Line 102-103: Consider rephrasing the two survival categories. While distinguishing cases with <1 week survival (worst prognosis or acute death) from those that survived longer than 1 week is meaningful, the label "chronic disease" is not an appropriate term and may mislead readers.
Line 149-154: The authors may consider consulting a medical statistician to perform a survival analysis for the study.
Table 2: It is unclear what the "Age" in the first row represents. Please clarify its definition.

Author Response
Dear Reviewer,
Thank you very much for taking the time to review this manuscript. Please find the detailed responses below and the corresponding revisions/corrections highlighted red in the re-submitted files.
Manuscript ID: animals-3476636-peer-review-v1 entitled "The pathological and histopathological findings in cats with clinically recognised hypertrophic cardiomyopathy are related to the severity of symptoms and disease duration".
This manuscript describes the pathological and histopathological findings in cats with clinical hypertrophic cardiomyopathy. I appreciate the authors for conducting this study, and I believe the information provided in their work contributes to the professional knowledge of feline cardiomyopathy. Here are some suggestions and comments for the authors to consider:
Thank you for that comment.
Line 91-94: Regarding the exclusion criteria, could you please explain the reasoning behind excluding cases that had received corticosteroid administration? Additionally, if a cat had chronic kidney disease but no systemic hypertension, would it also have been excluded? Please clarify the concern behind this criterion (perhaps in the discussion section).
Thank you for pointing out this concerns. The initial inclusion of long-term corticosteroid administration to exclusion criteria was based on the finding that in humans long-term corticosteroid administration is associated with hypertension (DOI: 10.1016/j.clinthera.2017.09.011). We found similar possibility in dogs (DOI: 10.1111/jvim.15331) but no information about cats and possible induction of hypertension due to steroids administration. We revised our database again and found no cat that would: have a history of long-term steroid administration, have HCM diagnosis and have no other excluding comorbidity. It allowed us to remove this confusing exclusion criterium and not change the study group.
Regarding chronic kidney disease, although hypertension in CKD is not always present, recent study by Flora et al. (DOI: 10.1016/j.jcpa.2024.11.006) showed that in old cats with myocardial degenerative changes and fibrosis, CKD was present regardless of the presence of hypertension. Although it may result in excluding too many animals, we used this exclusion criterium to characterise the pathomorphological picture of a feline heart with hypertrophic cardiomyopathy and without comorbidities. We added information in the discussion section (lines 354-364).
Line 95, Line 97, Line 98 (and throughout the manuscript): Replace symptom with clinical signs to align with standard veterinary terminology.
Thank you for that comment, we made changes throughout the manuscript and in the supplementary files.
Line 96: What does ‘auscultation disorders’ mean? Please clarify this term.
We add clarification in the text.
Line 96 (and throughout the manuscript): Replace dyspnea with respiratory distress for proper veterinary terminology.
We made corrections, as suggested.
Line 96 (and throughout the manuscript): Replace hydrothorax with pleural effusion
We made corrections, as suggested.
Line 96 (and throughout the manuscript): Replace hydropericardium with pericardial effusion
We made corrections, as suggested.
Line 99-100: If the duration data were retrieved from the period between echocardiographic diagnosis and death (including euthanasia), I suggest rephrasing "disease duration" to "survival time" for better clarity.
Thank you for that comment. We changed the term.
Line 102-103: Consider rephrasing the two survival categories. While distinguishing cases with <1 week survival (worst prognosis or acute death) from those that survived longer than 1 week is meaningful, the label "chronic disease" is not an appropriate term and may mislead readers.
Thank you for that comment. We rephrased the categories to be more proper (lines 101-103) and changed them throughout the text.
Line 149-154: The authors may consider consulting a medical statistician to perform a survival analysis for the study.
Thank you for that comment. We consulted a medical statistician during the initial analysis of the results. Unfortunately, the information about survival were not precise, i.e. an information “two years survival” could be either 24, 28 or 30 months. It was one of the reasons why we categorised patients into two groups (short survival / long survival). And it was the reason why a proper survival analysis was not possible.
Table 2: It is unclear what the "Age" in the first row represents. Please clarify its definition.
Thank you for that comment. We changed the term in Table 2 to be more accurate.
Round 2
Reviewer 3 Report
Comments and Suggestions for Authors
I appreciate the authors for revising this manuscript. I believe the information provided in their work contributes to the professional knowledge of feline cardiomyopathy.